# The Knowledge and Perceptions of Florida Pharmacists in Administering Inactivated Influenza Vaccines to Pregnant Women

**DOI:** 10.3390/pharmacy9020083

**Published:** 2021-04-16

**Authors:** Oluyemisi Falope, Cheryl Vamos, Ricardo Izurieta, Ellen Daley, Russell S. Kirby

**Affiliations:** 1Merck & Co., Inc., Kenilworth, NJ 07033, USA; 2College of Public Health, University of South Florida, Tampa, FL 33612, USA; cvamos@usf.edu (C.V.); ricardoi@usf.edu (R.I.); edaley@usf.edu (E.D.); kirbyr@usf.edu (R.S.K.)

**Keywords:** flu vaccination, pharmacy, perinatal care

## Abstract

Background: Influenza vaccine rates in pregnant women remain suboptimal despite the recommendations from healthcare organizations. Though pharmacists can provide immunization services as a result of the standing order, few studies have examined the role of the pharmacist in providing immunization to pregnant women or explored their perspective on their role in providing influenza vaccines among pregnant women. Purpose: This study explored the perceptions and knowledge of Florida pharmacists in administering inactivated influenza vaccines (IIV) to pregnant women. Methods: Semi-structured in-depth interviews guided by the theory of planned behavior were conducted with 18 licensed Florida pharmacists, including clinical and retail pharmacists. A thematic analysis was conducted. Results: The majority of pharmacists (94%) were knowledgeable about the IIV in pregnant women. Participants expressed mixed attitudes, identified barriers and facilitators, and subjective norms influencing vaccine administration in pregnant women. Participants expressed the importance of trust and how that influenced vaccine uptake. Participants also expressed their position not to only provide immunization services but also to counsel and educate patients. Conclusion: There is a need to strengthen immunization services, provided by pharmacists to more individuals, including high-risk groups such as pregnant women.

## 1. Introduction

During pregnancy immunity is lowered, making women susceptible to various infections, including the influenza virus illness [1]. In between influenza epidemics and during influenza pandemics, influenza illness in pregnant women can cause pneumonia, spontaneous abortions, preterm births, birth defects, and fetal loss [2,3]. There is also increased mortality in pregnant women during influenza epidemics; thereby, placing pregnant women as a priority group for receiving the inactivated influenza vaccine (IIV) [2,3,4,5]. The American College of Obstetricians and Gynecologists (ACOG) and the Centers for Disease Control (CDC) recommend that pregnant women should obtain the IIV [5,6,7]. Despite recommendations, the IIV uptake rate in this population remains suboptimal, with about 50% of pregnant women receiving the IIV yearly in the United States [8,9,10]. Barriers and reasons for influenza non-vaccination in pregnant women include safety concerns and first or second-trimester pregnancy [11,12,13]. Other barriers include lack of access, time, transportation, and non-availability of IIV at providers’ offices [12,14,15]. Provider barriers for influenza non-vaccination in pregnant women include insufficient time during visits, vaccine shortage, reimbursement challenges, lack of knowledge, and lack of enthusiasm to recommend the vaccine [14,15,16,17].

In the year 2000, the CDC recommended standing order programs, which authorized pharmacists and nurses to administer vaccines to adults [18]. This recommendation followed studies that demonstrated improved vaccination rates with these healthcare professionals [18]. Implementation of standing orders took the pressure off physicians and increased the vaccine coverage [18,19]. Vaccines provided through standing orders can be obtained at outpatient and inpatient facilities, pharmacies, assisted living facilities, home healthcare facilities, adult workplaces, correctional facilities, managed care organizations, and client, employee, and resident populations [18]. Among healthcare providers, pharmacists are the most accessible, as approximately 250 million individuals make a trip to the pharmacy every week [19]. Some of the barriers in the IIV uptake in pregnant women such as access issues, lack of time, transportation issues, and non-availability of vaccine at physicians’ offices can be overcome through community pharmacists as they are accessible and have extended hours and days of work, and usually have more vaccine stock [19,20].

Few studies qualitatively explored the role of pharmacists in administering vaccines. A quantitative study by Dolan et al. (2012) assessed pharmacists’ knowledge attitudes, and practices in treating and vaccinating pregnant women during the 2009 influenza virus pandemic. They reported that pharmacists did not believe they had an important role in administering IIV to pregnant women [21]. The state of Florida had a 21.6% influenza vaccine coverage among pregnant women during the 2009–2010 influenza season, and it also ranked the lowest among the states studied. [22]. The state of Florida would benefit from this research, as it hopes to create more opportunities and avenues for pregnant women to receive the inactivated influenza vaccine; emphasizing the role of pharmacists could play may help increase the vaccine uptake. This study aimed to explore the perceptions and knowledge of pharmacists in IIV administration to pregnant women, using the theory of planned behavior (TPB) as a guiding framework. TPB is an intrapersonal level theory with three constructs (behavioral attitudes, social norms, and perceived behavioral control), which all determine why an individual may or may not engage in behavior [23]. For this study, the theory was modified to include knowledge as a construct (Table 1). This was to account for the subjective norm, which was considered to be a weak predictor of intention [24]; therefore, it was important to understand what the individuals’ knowledge was and how it affected their intention, in this case, providing IIV to pregnant women.

## 2. Methods

This qualitative research study employed the use of in-depth interviews. Using the TPB framework, the interview guide asked questions about their knowledge, attitudes, subjective norms, and perceived control regarding administering the IIV to pregnant women. The interview guide was piloted. Licensed pharmacists in the state of Florida were chosen as the target population and were recruited via purposive and snowballing sampling. The exclusion criterium was pharmacists not being licensed in the state of Florida. The participants were stratified by type of pharmacy where they work (clinical/hospital setting or community setting). This stratification provided some variability to the data to assess the impact of the type of pharmacy where pharmacists work on their knowledge, attitudes, subjective norms, and perceived control related to administering the influenza vaccine to pregnant women. A list of sites where pharmacists work was generated, and the gatekeeper, or whoever was in charge at the various sites, was contacted to help gain access and recruit participants. Emails and flyers, which briefly introduced the study and described how to contact the researcher, were sent to licensed pharmacists to recruit study participants. The recruitment process for the licensed pharmacy interviews was achieved over a seven-month period between February and August 2019. There were a lot of cancellations after initially agreeing to participate in the study due to busy work schedules. However, to help assist the recruitment process for the pharmacist, a more vigorous recruitment strategy was employed, which involved approaching pharmacy directors to assist in the recruitment. Through the second round of recruitment, pharmacy directors themselves participated in the study, and the sample size of 16 was achieved upon saturation, as no additional information, new themes, or codes emerged. Two more pharmacists were interviewed to reach a total sample size of 18. The study was approved by the University of South Florida Institutional Review Board (IRB).

### Data Analysis

Recorded interviews were transcribed, and a distinctive participant identifying number was used to differentiate each participant. Transcripts were uploaded into MAXQDA 20.3 software (VERBI gmbh, Berlin, Germany) which was used to analyze and manage data. An a priori codebook was developed that had themes based on the interview guides. Analysis was carried out concurrently as interviews were being conducted and transcribed, and emergent codes were added to the codebook, with revised definitions as needed, and then finalized. The codebook was imported into the MAXQDA software and was applied to the relevant parts of the transcripts. An analysis was carried out by two independent coders. The primary investigator (OF) coded all the (18) transcripts, and a second coder (LB) coded 10% (2) of the transcripts independently to produce an inter-rater reliability measure. The Cohen’s kappa inter-rater reliability for this study was 84%.

## 3. Results

### 3.1. Participant’s Characteristics

Fifty percent of the participants were female, 44% were aged between 31 and 40 years, and 72% were certified to immunize. Seventy-two percent worked in community pharmacies, and 28% worked in a hospital setting. About 86% had the IIV in stock during the past influenza season, and only 6% experienced a back order for the IIV during the past influenza season (Table 2).

### 3.2. TPB Constructs Influencing IIV in Pregnant Women’s Practices

#### 3.2.1. Knowledge

When asked about their general knowledge (Table 3) on influenza vaccines, all (18) participants were knowledgeable about the influenza vaccine and its use to prevent the influenza virus illness. One participant stressed the importance of getting the influenza vaccine regardless of the side effects. Thirty-nine percent (7) of the participants also mentioned other things they knew about the influenza vaccines, such as the recommended groups for the vaccines. When asked about the efficacy of the vaccine, all participants said the vaccine was never 100% effective against the influenza virus, and it varied from year to year but encouraged the uptake as it may still confer some protection. However, few (2) felt the vaccines were not effective against the influenza virus disease based on their personal beliefs (Table 3).

When asked about IIV and pregnant women, all the participants knew about the benefits for pregnant women. Some of the participants (5) mentioned the contraindication of live vaccines in pregnant women. Some of the participants (5) mentioned the IIV could be received at any trimester by pregnant women, while (3) mentioning not being sure what trimester was safe for pregnant women to obtain the vaccine. When asked about the benefits of the IIV during pregnancy, most (13) respondents mentioned the vaccine provides dual protection for the mom and baby from the influenza illness, keeping them both healthy (Table 3).

#### 3.2.2. Attitudes

Most participants (14) believed it was advantageous to the community and pregnant women that pharmacists provide immunization services. Participants described positive attitudes expressed in the following sub-themes: (a) accessibility, (b) experts, (c) increased scope, and (d) ease for practices. Participants believe they are the most accessible healthcare providers (Table 3). Participants (6) believed they were experts and knowledgeable; hence, that placed them in a great position to provide immunization services to pregnant women. Participants (8) expressed that providing IIV to pregnant women increased the scope of their practice. Participants (7) expressed the ease for women and practices/providers by taking off workload when pharmacists provide the vaccine. Participants expressed less positive attitudes towards providing IIV to pregnant women, such as liability issues (9), lack of training or education (1). One participant, however, was not sure pharmacists were allowed to immunize pregnant women (Table 3).

#### 3.2.3. Perceived Control

When asked about facilitating factors to provide IIV in pregnant women, participants (9) described the need to employ other staff such as pharmacy technicians to assist with the workload, so they could focus on immunization services when the need arises (Table 3). A few participants (6) felt outreaches and campaigns tailored to pregnant women may facilitate pharmacists providing the IIV to the pregnant women as they would create awareness. Two participants reported that, with support and education, pharmacists would be more likely to provide immunization services to pregnant women. Participants (5) mentioned that when an incentive was provided to the pregnant women to obtain the IIV, and when it was provided to the pharmacist to provide the vaccine, this could facilitate the provision of the vaccine (Table 3). Figure 1 shows a summary of the facilitators expressed by participants comparing the types of pharmacy (clinical vs. community) where participants worked (Figure 1).

When discussing the barriers, pharmacists expressed various sentiments, including lack of access to pregnant women, increased workload, workplace environment, and lack of education for themselves as pharmacists regarding providing vaccines to pregnant women.

Other participants (6) expressed that not having access to pregnant women may be a barrier. Thirteen participants believed that providing immunization services increased their workload. Other participants (2), however, did not feel providing immunization services to pregnant women interrupted or increased their workload. A few participants (3) shared the sentiment that the pharmacy environment was not in itself conducive to provide immunization (Table 3).

Other themes regarding factors that may affect pharmacists’ volitional control in administering the IIV to pregnant women include the type of insurance or the community settings (rural, urban) of the pregnant woman. When asked about the insurance type, all licensed pharmacists expressed sentiments that insurance type may be a barrier or a facilitator depending on the type (Table 3). Some of the participants (9) believed there was less access to healthcare services in rural areas. However, one participant felt individuals in rural areas might have access to immunization services because of their relationship with their pharmacist. Others (3) believed that barriers to immunization in any region had to do with the patient’s socio-demographic characteristics or vaccine stance (Table 3).

#### 3.2.4. Subjective Norms

Participants were asked about the influence of the following in regard to providing IIV to pregnant women: (a) the CDC, (b) the APhA (American Pharmacist Association), (c) the Florida state statutes, (d) physicians, (e) pregnant women, and (f) their peers. All licensed pharmacists believed the CDC served to play the role of quality control and provided the vaccine guidelines regarding the administration to pregnant women and the general public (Table 3). Participants (11) mentioned the APhA to be an important resource through which they obtained their training on immunization and for continuing education. They also expressed that the APhA acted as an advocate for pharmacists to become immunizers, while a few (*n* 4) mentioned that the APhA provided information and guidelines regarding immunization in general, but not specific for pregnant women (Table 3). When asked about the influence of the Florida state law regarding pharmacists providing immunization services under the supervision of a physician within a framework of an established protocol [25], 8 participants expressed positive feelings about it. A few participants (3) expressed neutral feelings about the standing order. Other (6) participants expressed negative feelings about the standing order and did not believe it was needed. A few (3) of the participants mentioned the standing order might not be good for independent pharmacies, as it was costly. The two independent pharmacists interviewed were not immunization certified (Table 3).

All participants believed that physicians play a major role with regards to referring the women and creating awareness that they can get their shots at the pharmacy. One participant mentioned that physicians were key to increasing the number of women getting immunized. Others (3) believed that physicians might also be a barrier for pregnant women coming to the pharmacy to receive the IIV (Table 3). Participants were neutral about their colleagues choosing to or not to provide immunization services to these women; one participant, however, talked about how that impacts the vaccine uptake in pregnant women. Another participant mentioned that the older pharmacists might not be willing to provide immunization services to pregnant women. One participant mentioned if more pharmacists were getting certified to immunize, it might spur others up to want to get theirs (Table 3).

When asked about the influence of pregnant women on pharmacists administering the IIV to them, pharmacists expressed that providing IIV services to pregnant women would not be difficult due to the fact that patients trust pharmacists with their health questions. This sentiment was shared by 2 participants. However, two other participants shared the sentiment that pregnant women might trust a doctor over a pharmacist. Among the 11 pharmacists who had never immunized a pregnant woman before, 8 reported that they would be willing to provide immunization in the future, while 3 reported they were not willing to do that (Table 3). Among the 7 who had provided the vaccine to pregnant women before, one participant mentioned providing the vaccine was a basic skill, and another participant mentioned it was not their favorite activity (Table 3). Pharmacists in clinical settings, regardless of immunization certification status, did not provide immunization in clinics. Clinical pharmacists reported nurses as the individuals responsible for administering immunization in hospitals.

## 4. Discussion

When discussing the pharmacist’s knowledge about the influenza vaccine, generally, and its benefit to pregnant women, they were conversant with the general influenza knowledge including the types, recommendations, efficacy, and side effects; and also the knowledge specific to pregnant women, such as the safety, the type of vaccine used, and the benefits. However, only about 25% of them mentioned that the IIV could be received by pregnant women at any trimester. This was in line with the ACOG recommendations for influenza vaccines to be administered at any trimester [5,6,7]. About 72% of the pharmacist also reported knowledge of the dual benefit of protection of the mother and infant for the first six months of life [6,26]. Regarding the attitudes towards administering IIV to pregnant women, participants expressed overall positive sentiments. They expressed being accessible, this was consistent with what was reported in the literature [27]. Participants viewed themselves as knowledgeable and experts to provide counseling and IIV to pregnant women, and providing such services increased their scope of practice. The study by Dolan et al. (2012) also reported that providing IIV was viewed by pharmacists to expand their scope of practice [21]. In terms of providing ease for patients, research had proven that pharmacists providing immunization services were cost-effective [20,28]. However, participants expressed negative feelings such as liability and not having enough information on the pregnant women, consistent with the literature [21].

For perceived control, participants mentioned facilitating conditions to include provider recommendations, employing more staff (pharmacy technicians), educating pharmacists, and marketing campaigns. Higher IIV rates were seen among pregnant women who received a recommendation from their provider when compared to those who did not, also pharmacy technicians were known to aid pharmacists to engage in a more effective workflow. Campaigns and outreaches were also associated with increased immunization rates [20,29,30,31]. Barriers included an increased workload, no access to pregnant women, pharmacy not being perceived as a conducive environment, lack or types of insurance, and socio-demographic factors. Studies showed these to be true [21,27,32,33]. However, not all participants agreed that providing immunization to pregnant women increased the workload. Regarding subjective norms, pharmacists expressed that the CDC served as a resource for quality control and updated vaccine information, the APhA was viewed as a training, certification, and continuing education resource. Studies showed that pharmacists were okay with providing immunization services if they got information from the required organization [21]. In discussing the influence of the Florida standing order, participants reported varied views: some felt it was needed, some felt neutral, and others felt it was not necessary. Since the inception of the standing orders for immunization, vaccine uptake rates have increased [19,34,35]. Participants perceived physicians to play a key role in educating and referring women to pharmacies for immunization [29,30]. Additionally, pharmacists expressed that pregnant women would rather go to their Obstericians /Gynecologists than come to them. A study by Wang et al. (2014) reported that pharmacists were the most trustworthy healthcare providers next to physicians [36].

Overall, the pharmacists expressed sentiments that they were equipped to provide immunization services and counseling to pregnant women; 17%, however, reported that they were not willing to provide immunization services to these women. Participants also called on the need for the standing order to be reviewed to allow for independent pharmacists to provide immunization services. A study by Burson et al. (2016) showed that independent pharmacists were less likely to be certified [37]. This was also demonstrated in the study, as the independent pharmacists interviewed were not certified to immunize.

There were several study limitations. First, it was a qualitative study with a small sample; therefore, it cannot be generalized. Another limitation was that there might be a social desirability bias from the pharmacists. Last, the study did not include the perspectives of pregnant women and providers, such as the OB/GYN, midwives, or family medicine doctors, who usually attend to pregnant women on pharmacists administering the influenza vaccine to pregnant women. This study had several strengths: The qualitative nature of this study was important for exploring how licensed pharmacists perceived themselves and their role in providing immunization to pregnant women. In addition, the study was guided by the theory of the planned behavior theoretical framework. Based on the qualitative results of this study, future research could be developed using quantitative methods to reach a wider and more generalizable audience.

## 5. Conclusions

Vaccine uptake was optimal in situations where vaccines were available, healthcare providers recommend the vaccine, and individuals accepted the vaccine [30]. While this study did not assess the acceptance by the pregnant women or recommendation by providers, it assessed IIV availability through pharmacists. The authorization of pharmacists to provide immunization through the standing order was proven to increase the overall immunization rates in adults [38]. The standing orders, however, did not specify pregnant women, a specification of this group might help to increase the influenza vaccination. Further, there should be considerations for the autonomy of pharmacists to be able to be considered as practitioners being able to provide immunization independent of a physician, as stated in the standing order [25]. Future studies could apply this study to understand the pharmacy’s role in administering vaccines to other high-risk groups such as children and during outbreaks. Pharmacists play an important role in increasing IIV rates among pregnant women inadvertently providing coverage for moms and infants. Pharmacists were providing immunization to adults for almost 20 years now and are capable of providing immunization to pregnant women. Clinical practice should be driven by research and not personal beliefs or views; also, pharmacists providing influenza immunization should not be diminished due to lack of knowledge. Continuing education training and certifications should include strategies targeting pregnant women and other at-risk groups in their vaccine certification training.

## Figures and Tables

**Figure 1 pharmacy-09-00083-f001:**
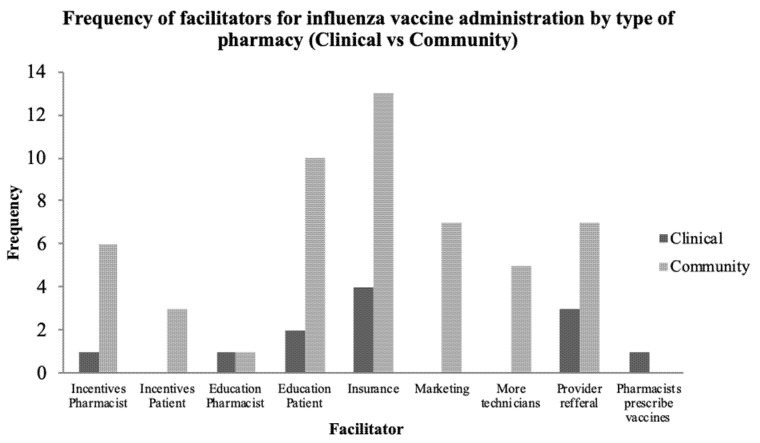
Frequency of facilitators for influenza vaccine administration by type of pharmacy (clinical vs. community).

**Table 1 pharmacy-09-00083-t001:** Modified TPB model constructs as applied to study.

Construct and Subconstructs	Definition	Application
Attitude i. behavioral belief ii. evaluation	A person’s belief regarding performing a behavior The notion that doing a particular thing has specific outcomes The value associated with the behavioral outcome	Information on what pharmacist feel about how beneficial administering influenza vaccines to pregnant women is, and how comfortable they feel regarding administering the influenza vaccines to pregnant women (their favorable/unfavorable disposition towards doing the behavior)
Subjective norms i. normative beliefs ii. motivation to comply	Notions of how certain individuals view and approve, or disapprove of performing a behavior A person’s engagement in a particular action if they feel people close to them think they should Motivation to engage in practice based on what the individual thinks	Information on how other healthcare providers and pregnant women’s attitudes towards pharmacists can influence administering influenza vaccines to pregnant women
Perceived control	How much a person feels that engaging or not engaging in a behavior is under their volitional control	Information on how much control pharmacists think they have regarding administering influenza vaccines to pregnant women
Knowledge	Theoretical or practical understanding of a subject	Information on the pharmacist’s general knowledge of influenza vaccines, especially as it relates to pregnant women

Note: Constructs of TPB are derived from Glanz, K.; Rimer, B.K.; Viswanath, K. *Health behavior and health education: theory, research, and practice.* John Wiley & Sons; 2008.

**Table 2 pharmacy-09-00083-t002:** Licensed Florida pharmacists’ demographics.

Characteristics	Number	Percentage (%)
Certified to immunize		
Yes	13	72
No	5	28
Age range		
18–30	7	39
31–40	8	44
>51	3	17
Sex		
Female	9	50
Male	9	50
Years of practice		
0–5	5	28
6–10	6	33
>10	7	39
Location of pharmacy		
Rural	1	06
Urban	12	66
Sub-urban	5	28
Influenza vaccine in stock		
Yes	14	78
No	2	11
Not applicable	2	11
Influenza vaccine back order		
Yes	1	06
No	13	72
Not applicable	4	22
Ever immunized pregnant woman		
Yes	7	39
No	6	33
Not applicable	5	28

Not applicable: pharmacists either do not stock vaccines in the facility and/or experience back order and are not certified to offer immunization.

**Table 3 pharmacy-09-00083-t003:** Theory of planned behavior (TPB) major themes and subthemes and representative quotes.

Themes	Representative Quote
**Knowledge**	
General knowledge	“It is very important that you get it every flu season, it is not going to make you get flu… Even if you have an egg allergy, you are still able to get the flu vaccine, the incidence of people who actually have a true reaction or true allergy to flu vaccine content is very minimal, it is rare. And there is another type of vaccine called the flublok that we give or recommend for you, that will be appropriate if you were concerned about that.” ~Pharmacist 16
	“It’s very safe. There isn’t a whole lot of things that are safer than flu vaccine. When patients ask me how safe the flu vaccine is, I explain to them that the CDC, the FDA, whatever government agencies…, whatever I think they can understand, says that I can give the Flu vaccine to anybody as young as 6 months of age; if it’s that safe, it’s safe for everybody. The only contraindication to flu vaccine is if you are allergic to eggs.” ~Pharmacist 10
Recommended groups	“If you are over the age of 6 months you are recommended by the CDC and most of other guidelines to get it. It’s a respiratory virus that can be transmitted. It can cause from mild to severe disease, especially in our high-risk patients need to be vaccinated.” ~Pharmacist 8 “The vaccine does not always protect from the virus a 100%, but it provides some sort of protection, immunity. I think it helps especially in pregnancy and other special populations it is something people should get. In general people should get the vaccine even if it is not 100%.” ~Pharmacist 7
Efficacy	“I would say that I believe it doesn’t really stop you from getting the flu, because sometimes it gives you the symptoms of the flu after you get the flu shot.” ~Pharmacist 17 “I got vaccinated in school and after a day or two I was really sick… It was like I was going to die. So, it was during finals week. I almost failed school, I couldn’t explain why the vaccine I got me sick immediately, ever since, I didn’t take the vaccine.” ~Pharmacy 9 “In my estimation it is 50–50. It does not hurt to get the vaccine, it does not necessarily mean if you get the vaccine you won’t get the flu, but it does provide some sort of protection, it does not hurt to get it, it is something I will recommend, and I will get it myself.” ~Pharmacist 7
IIV and pregnancy knowledge	“In most pregnant women the only contraindication usually is the live vaccine, to not administer it to pregnant women, but the inactivated vaccine is safe in women who are pregnant and women who are breastfeeding.” ~Pharmacist 2 “We are able to immunize pregnant women, but we use the preservative free.” ~Pharmacist 15 “We try to give them flu shot and tetanus shot and I know that they can get it at any time during their pregnancy, so whatever the trimester they are in, as long as they are within the flu season, we try to give it to all of them in our pharmacy setting. We do have the standing order with the doctor so we can administer it. If they are 19 years and older, we can give it to them.” ~Pharmacy 12 “It is safe in pregnant women. They definitely should receive it, and the vaccine does not affect the child too, so it provides a double coverage to the mom and baby…” ~Pharmacist 11
**Attitudes**	
Access	“Biggest advantage is accessibility. Can I pick up my medication? sure. Can I get counseled? Sure! Oh, while I am here can I get my flu shot? Sure! So one stop shop for majority of all their needs.” ~Pharmacist 6
Experts	“I think that a lot of people lack the knowledge. I think that they feel that the vaccines are going to make them sick. We know a lot; we interact with the CDC a lot so we can explain to them the importance of it versus the risks of not having it. So, I think that we are just that one extra level of education for them to feel safe about getting the vaccines.” ~Pharmacist 8 “We are very knowledgeable, and I think it’s just now that our knowledge is starting to come into play, we have been doing this forever…” ~Pharmacist 6 “…we are fully educated on immunization as doctors are, and most the patients get are side effects or allergic reactions to the vaccine.” ~Pharmacist 17
Increased Scope	“I also think that it’s a good way for people to see the advancing role of a Pharmacist; that they are not just counting pills, but they have a lot more roles.” ~Pharmacist 8 “…it increases our scope of practice as practitioners.” ~Pharmacist 2
Ease for practices and patients	“It saves them (pregnant women) time and money. More than likely if you are in the pharmacy, just speaking of pharmacies now, most pharmacies have convenience stores, so you are doing shopping there anywhere, to rephrase my earlier statement, it’s a one stop shop. No appointments needed, no doctors, just let us know when you want it, and we will take care of it for you.” ~Pharmacist 6 “It helps out some of those strained medical facilities that may be inundated with pregnant women…, I think we, on their end, we are a big help to doctors. A lot of doctors’ offices in particular run out of the vaccine very quickly, whereas a lot of the larger chains keep it in stock.” ~Pharmacists 2
Limited knowledge	“I think the only disadvantage is just the pharmacist him or herself, maybe not feeling confident in their skills or maybe education wise, if they haven’t done it in a while, they are not used to doing it so often or not knowing the type of patient that is in front of them, whether vaccine live vs. attenuated which is okay you know. Or you know maybe education wise, they might not feel confident, other than that, no.” ~Pharmacist 3 “I think they have to go get it (IIV) at their primary physician. I don’t think that pharmacists are allowed to do that (provide IIV to pregnant women).” ~Pharmacist 4
Liability	“There’s room for litigation, as the pharmacists may not be able to respond to potential complications if they arise, though rare. Liability as a result of an inability to respond to complications we cannot anticipate.” ~Pharmacist 14
**Perceived Control Facilitators**	
More technicians	“Another thing again is manpower, because a technician can help while you will be able to provide these services. They can take away some of the manual aspect of the work, while you do the clinical aspect of the work.” ~Pharmacist 7
Marketing	“There are some promotions that go on every flu season, we do like a blog entry, just like promotional items we have with our preferred pharmacies to encourage customers to get the flu vaccine. It is not targeted to pregnant women specifically but can be.” ~Pharmacist 5 “When they say “get your flu shot” it’s like a general thing but if they can just put another caption to let pregnant women know … So, I mean everything starts from that awareness, let pregnant women know even from the hospitals, commercials though the TV, everywhere… that it is safe, if there is a lot of awareness, they will feel more comfortable walking in.” ~Pharmacist 18
Education	“I think educating us, providing more education around immunizing pregnant women specifically. I think as long as the pharmacists feel like they are supported, they know it is recommended by the CDC then they will be more likely to provide the immunization.” ~Pharmacist 15
Incentives	“We get incentivized on the back end, so it’s more like our flu goals. If you have met your goals, and your store is up to par for other things, you can ask for a raise…, we immunized anyone we could find with a pulse, cos we had to meet our goals… But patients also get the incentives, and that is what helps them come to us. I work at Xxx typically, and they get a $10 coupon.” ~Pharmacist 13
Insurance	“If the (insurance) pay for it, we are going to have an increased influx of women wanting to get it; if patients have to pay out of pocket then we have to think about costs. HMOs (Health maintainance Organizationa) are going to be the ones that dictate where they go. So those are going to be the ones that probably won’t pay for vaccine at the pharmacy versus telling them they have to go to their primary care physician. For like PPOs (Preferred Provider Organization) and so forth, the patients usually get autonomy and get to pick where they go.” ~Pharmacist 13
Community settings	“Rural is helpful, pharmacists have a relationship with their patients, they are slower paced, you can actually identify them because you know them. You can say, ‘hey Mrs. Xxx we think you should get this vaccine today’, and that is a better way compared to urban where you don’t really have that relationship with your patients, who are more in an urban area.” ~Pharmacist 13
**Perceived Control Barriers**	
Access to pregnant women	“Most pregnant women think the doctor has to do it, or they have to go to the clinic. I think the lack of knowledge about using pharmacists is the biggest barrier.” ~Pharmacist 11 “The pharmacist may not have full comprehensive health history on the patient because the pharmacy system they are not as connected to the hospital system, I am speaking more for community pharmacies…” ~Pharmacist 16
Increase workload	“It is added work for the pharmacist, it is a lot more work. In my experience when I was in retail, in fact one of the reasons why I left retail… On a given day I have 5–10 immunizations, with people waiting for me, doctors call back, counseling sessions. The patient load on the pharmacist is a lot, and now some retail store requires the pharmacist to give a number of vaccinations in a day or a month, it’s your job. It should not be a quota thing for pharmacists to provide such services.” ~Pharmacist 7 “You can have your technician set your workflow, set it up. So that way, once they identify a patient who needs to be vaccinated, the pharmacists comes in, vaccinates the patient, asks questions, answer questions, counsel the patient, do your mtm (medication theraphy management), it won’t take you long, you just have to have that built into your workflow.” ~Pharmacist 3
Pharmacy environment	“I think there would be some discomfort, because the pharmacy is not an intimate setting, and pregnancy is very tough. Because of the setting, pharmacies usually have an open concept that would probably prevent them from freely going to get retail immunization, you know, the vaccine. It is not pregnancy friendly” ~Pharmacist 11
Community settings	“I think that depends more on the demographics…, the patient’s education level, their level of awareness, exposure. We already have a propaganda by people who are against any sort of vaccination, so that’s another thing. It all depends on the demographics, and social standing, education status all those things. I work in a rural area of Florida, and I have seen that healthcare seeking behavior has to do more with the patients sociodemographic.” ~Pharmacist 14
**Subjective Norms**	
CDC	“… they do a lot of the monitoring, and quality checks and you know quality control just to make sure it is the right vaccine for coverage” ~Pharmacist 14
APhA	“APhA (American Pharmacist Association) gives you the continuing education credits. They have the thing every year where they talk about the flu vaccine, the hits and the misses, all that fun stuff, information.” ~Pharmacist 13 “Generally, APhA doesn’t directly affect vaccines for pregnant women…” ~Pharmacist 1
Florida State	“It has not limited us in any way. My belief is that it was a necessary step in order to achieve future autonomy, because we had to show how we impact immunization rates and how we can do it safely. I don’t think we would have gotten autonomy off the back from a legislative standpoint. We have to earn that.” ~Pharmacist 15 “I am kind of neutral about it. I mean everything is under the supervision of a doctor, regardless if it is the influenza vaccine or prescription for any medication.” ~Pharmacist 17 “I actually think it is pointless, the physician is just on a piece of paper, because if there was an emergency going on with like you know, vaccine reaction or something like that, I am not going to call the physician, I am going to call 911 to come get the patient. I don’t think the physician is necessary.” ~Pharmacist 16 “…if you want to provide immunization services, you need to do that if the cost is worth it. While this has expanded the opportunity for pharmacists to immunize, nothing has really changed for us independents.” ~Pharmacist 14
Physicians	“…if the physicians refer the patients to the pharmacy, you know, go to your local pharmacy and get vaccinated. If they can encourage that behavior that would be the key to changing the number of pregnant women getting vaccinated.” ~Pharmacist 12 “I think it might present a conflict of financial interest for the physician, it’s like sending your patient away to somewhere else where you will not be able to recoup funds.” ~Pharmacist 16
Pregnant Women	“… a lot of people trust the pharmacist’s knowledge of the drug, they trust the pharmacists of drug interactions, there’s a lot of trust in pharmacists in what they provide. As long as they are certified, I don’t think pregnant women will have any reservations getting the vaccine from the pharmacist” ~Pharmacist 17 I’ve always considered us as the first line because, with the patient, they come to us first and they have that trust-factor with regards to their health questions ~Pharmacist 9
Peers	“The older pharmacists won’t do it, they may be like, ‘why are you immunizing pregnant women? We don’t do that here’, they think they are a higher risk patient group for vaccination.” ~Pharmacist 13 “I think if a pharmacist has said no to a pregnant woman to administer the vaccine to them, they will probably be scared to go to any other pharmacist and that will have a big influence” ~Pharmacist 17 “…I mean, if everyone is doing it, if everyone is providing a service, then I am going to have to try to transition to that, just like when it comes to…,you know, things are advancing so you have to advance with it.” ~Pharmacist 14
**Practice**	“I mean, it isn’t brain surgery, or trying to run a catheter through someone’s femoral artery, it is just an IM (Intra muscular) procedure, it is really not a big deal. Eventually it felt like riding a bike, I just get up, stick the needle, and it is done… but you may want to assess the general state of the pregnant woman.” ~Pharmacist 7 “No, it is not something I am interested in.” ~Pharmacist 14 “Absolutely not! When I first started immunizing pregnant women gave me the most anxiety because they ask you all the questions about like mercury, autism, all the things… The lot number, the manufacturer of the vaccine, then it makes you kind of question what you are giving them… They are not my favorite population to vaccinate.” ~Pharmacist 3

## Data Availability

Data are available on request due to privacy restrictions. The data presented in this study are available on request from the corresponding author. The data are not publicly available due to IRB privacy restrictions.

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
