# Peer review of "The Knowledge and Perceptions of Florida Pharmacists in Administering Inactivated Influenza Vaccines to Pregnant Women"

_pharmacy, 2021, doi:10.3390/pharmacy9020083_

Round 1

Reviewer 1 Report

Dear Authors,

I have  enjoyed reading this manuscript. Please find few hints, I would like authors to adress in this paper.

Please describe the aim of your project. There is an anwser in the conclusion section, but  is not fully mentioned at the beggining. Highlight is needed. 

There is not study limitation section, which you could kindly provide, as well please describe patient recruiting process. 

Sample, is very small. I would add that further investigation is needed. 

I would be happy to review this article furhter after changes are done. 

Thank you.

Author Response

Thank you for reviewing our manuscript. Please find attached response comments.

Regards.

Reviewer 2 Report

Pharmacy Review Feb 2021

Abstract

  • Introduction
    • Line 32 define CDC
    • Line 47 – what are the barriers cited, perhaps list earlier in the introduction and propose how pharmacy access would address, is it accessibility? Insurance coverage? Other?
    • Line 53 – did or did not?
    • More information needed regarding Florida’s standing order, when implemented, any limitations, etc.

Methods

  • When was the study conducted, important in relation to the implementation of the standing order
  • How were the interviewed dispersed, electronic or paper
  • How recruited?
  • How long was the study open
  • Information categories asked
  • Any exclusion/inclusion criteria?

Results

  • Patient characteristics are mostly duplicative of the table. Choose one or the other to list result
  • Response rate?
    • Only 18 participants, unlikely representative of the state as a whole.

Discussion

  • Having trouble making the link between pharmacist knowledge and how this will translate into improved outcomes for patients, I’m sure that is some data on this but may need to be bolstered. This makes me think that the overall focus of the study, intro, title, etc should be changed to pharmacist knowledge and perceptions of the IIV in pregnant women?

Conclusion

Author Response

(The authors gave the same response as above.)

Reviewer 3 Report

Manuscript ID: pharmacy-1137564

Title: The Role of Pharmacists in Administering Inactivated Influenza Vaccines to Pregnant Women

This thematic analysis work is sound interesting. The authors addressed the problems persisting in the community regarding influenza vaccination among the pregnant women and discussed the licensed pharmacists’ role in more vaccination to the pregnant women since it is recommended by CDC standing order programs. However, there are several flaws need to address.

  1. In title, authors need add the study location since it does not represent the whole world or community.
  2. The purpose written in the abstract does not represent the title completely. Both should be aligned.
  3. Full form of IIV need to write in the first appearance within bracket and subsequently IIV.
  4. “Despite recommendations, IIV uptake rate in this population remain suboptimal with about 50% of pregnant women receiving the IIV yearly.” Does this prevalence represent the whole world? Authors always need to clearly mention any data belongs to which group to avoid misleading messages.
  5. Table 1 needs appropriate citation.
  6. “This qualitative research study employed the use of in-depth interviews.” Does this interview follow any previous method? Or Modified previous method? Authors can cite appropriately or if they designed it by them, also they can mention it.
  7. How authors calculated sample size did not mention. Why the sample size 16 was saturated sample size? Answering this is crucial for this study.
  8. “The study was approved by the University of South Florida Institutional Review Board (IRB)” Authors need add reference number of the approval.
  9. More comprehensive presentation of the results is required. In Table 2, what does it represent by NOT APPLICABLE?
  10. Author can prepare a Table where they can precisely present specific study category and key findings of participants response. For example,

No.

Item

Key response

1

Knowledge of IIV

Knowledgeable (100%)

2

Received IIV

Any trimester of pregnant woman (???)

Not sure (???)

  1. All quotation of response by respondents can be shared in the supplementary file. It seems duplication of the information in the main text.
  2. Discussion is well written. However, throughout the manuscript, minor grammatical error needs to amend.
  3. Some references are not well formatted. DOI number can be included, and website link did not provide where applicable, such as reference no. 6. Last access date should be provided where applicable.

Author Response

(The authors gave the same response as above.)

Round 2

Reviewer 1 Report

I accept all changes. No further modification are required.

Author Response

Thank you

Reviewer 2 Report

I commend the authors for their work on revising the methods, but considering the very small sample size, I cannot endorse this study. The results are interesting but do not mean anything to me as they cannot be generalized. I think this might be interesting if revised as an opinion paper or commentary but not as a research study.

Author Response

Thank you for your comment, we understand your reservation about the sample size. But it was a qualitative study, and the sample size was justified as saturation was reached. Also, a limitation of qualitative studies is that they are not generalizable.

Regards.